# Learning Robot Manipulation from Audio World Models

*Abstract*— **World models have demonstrated impressive performance on robotic learning tasks. Many such tasks inherently demand multimodal reasoning; for example, filling a bottle with water can make visual information alone ambiguous or incomplete, thereby requiring reasoning about the temporal evolution of audio, accounting for its underlying physical properties and pitch patterns. In this paper, we propose a generative latent flow matching model to anticipate future audio observations, enabling the system to reason about long-term consequences when integrated into a robot policy. We demonstrate the superior capabilities of our system through two manipulation tasks that require perceiving in-the-wild audio or music signals, compared to methods without future lookahead. We further emphasize that successful robot action learning for these tasks relies not merely on multi-modal input, but critically on the accurate prediction of future audio states that embody intrinsic rhythmic patterns.**

## I. INTRODUCTION

Recent developments in world modeling offer new possibilities for addressing challenges in robotic manipulation. Research in this domain has primarily concentrated on the following directions: 1) video-based models [1] that predict future visual frames from present observations, encoding the causal dependencies critical for physical interaction. Achieving high-fidelity visual prediction often necessitates large-scale generative models, which in turn impose considerable computational demands and latency. 2) To mitigate these limitations, recent state-of-the-art work explores latent vision-language world models [2] that learn internal representations of future states, thereby bypassing the need for explicit full-frame reconstruction. In this work, we explore another line of research, audio-driven world models, where audio provides rich information for robotic manipulation.

Existing multimodal robot learning approaches typically incorporate audio signals as an auxiliary sensory modality to enhance performance compared to vision-only policies, in tasks such as bagel flipping [3] or food scooping [4]. Nevertheless, in many scenarios, audio is not merely supplementary, as it can provide essential information for robot learning, particularly when visual cues are scarce or unclear [5]. For example, as shown in Fig. 1, water filling is an inherently multimodal task that strongly relies on auditory feedback due to liquid occlusion, transparency, and visual ambiguity, making it necessary to reason over the temporal evolution of audio cues, such as pitch patterns, to determine whether the bottle is full. Unlike action-free vision-based world models that can generate a wide variety of interactive environments [6], the audio cues in these tasks often reflect certain intrinsic physical dynamics. Consequently, predicting future audio observations is critical for informing robot action policies.

In this paper, we propose a generative model that predicts future audio states in conjunction with a robot action policy. Flow matching [7] has emerged as a promising alternative to diffusion models, offering faster and higher-quality sampling. As illustrated in Fig. 1, our framework employs a transformer-based flow matching approach in the latent space to efficiently generate temporally consistent audio representations. Across both simulation and real-world experiments, we demonstrate that leveraging audio predictions for robot action generation leads to superior performance compared to methods without future lookahead.

## II. RELATED WORK

Extensive literature has investigated the use of audio information in multimodal learning with visual data [3, 4, 8]–[10]. These frameworks frequently leverage audio signals as a complementary modality to refine policy execution beyond the capabilities of vision-centric models. Prior work in this domain has looked at using sound to improve contact-rich robot manipulation skills, including tasks such as bagel flipping, food scooping, granular material manipulation, etc. In these works, various representation learning frameworks have been applied to either explore the audio-visual correspondence as a form of cross-modal self-supervision from video with contrastive learning [11], or using a pre-trained implementation of foundation models for obtaining audio-visual representations [12]. Far from being secondary, auditory input can be indispensable for robotic training, especially in environments where visual sensors encounter interference or insufficiency. [13] has successfully tackled partially observed and heavily occluded manipulation tasks, such as taking a key out of a bag, leveraging sound and memory to resolve the partial observability. [14] has demonstrated that when vision is blocked, auditory data becomes the primary driver for successful sim-to-real transfer, particularly in complex fluid-handling tasks like pouring.

We draw inspiration from these works, tackling robot manipulation tasks that strongly rely on auditory feedback, such as water filling and piano playing tasks in Fig. 1. We transition from a reactive robot policy to an audio world model, enabling more robust reasoning over the temporal evolution of acoustic cues.

## III. METHODS

Fig. 1 presents the structure of our method. Given a source audio $\mathbf{s}^{-L':0} \in \mathbb{R}^{L' \times d_s}$ of length $L'$, our method generates a sequence of future audios $\mathbf{s}^{1:L}$ and the corresponding robot action chunk $\mathbf{a}$. Our proposed model consists of three phases. First, we pre-train an autoencoder for an expressive and

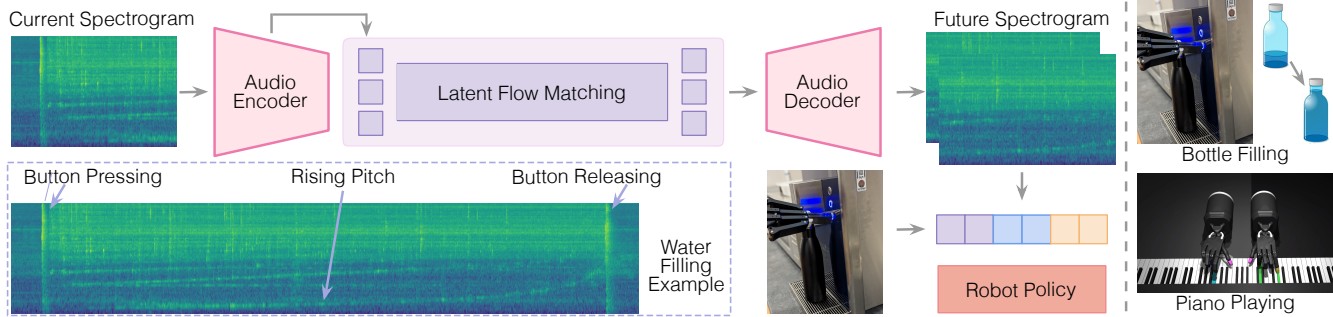

Fig. 1: Overview of the proposed method. The source audio is first encoded into a latent representation. Given the current audio segments, a flow-matching transformer estimates the generating vector field from noisy audio latents. This vector field is then used to solve the corresponding ODE, producing the future audio latents. The resulting sequence of future audio latents is decoded into audio spectrograms. Finally, a robot policy is trained using both the current and predicted future audio spectrograms along with image observations.

smooth audio latent space. Next, we employ flow matching as the backbone of our world model to generate a sequence of future audio latents, given the driving conditions. The generated future latents are decoded to the actual audios. Lastly, a policy model is used to predict the robot actions from the synthesized audio frames.

**Audio Latent Autoencoder** We represent audio using spectrograms, as they provide rich time–frequency information that effectively captures the signal's temporal and spectral features. For example, in the water-filling task, the spectrogram in Fig. 1 clearly indicates the button pressing and releasing actions, as well as the increasing pitch during the water-filling process. We train AudioMAE [6] to encode and decode the spectrogram with the reconstruction loss.

**Flow Matching in Latent Space** We employ flow matching [7] for the latent audio generation by reconstructing a target vector field computed from the corresponding audio latents. We regress on a vector field $\mathbf{v}_t(\mathbf{x}_t, \mathbf{c}_t; \theta)$ where $\mathbf{x}_t$ is the sample at the timestep $t \in [0, 1]$, and $\mathbf{c}_t$ represents driving conditions of consequent audio frames. We adopt the vector field predictor proposed in [15], which is modified from DiT [16], and decouples frame-wise conditioning from time-axis attention mechanism for temporally consistent latents generation.

For training, we choose audio latents $\mathbf{w}_{s^{1:L}}$, and construct the target vector field $\mathbf{u}_t(\mathbf{x}|\mathbf{w}_{s^{1:L}})$ with noisy input $\varphi_t(\mathbf{x}_0) = (1-t)\mathbf{x}_0 + t\mathbf{w}_{s^{1:L}}$ ($t \sim \mathcal{U}[0,1]$ and $\mathbf{x}_0 \sim \mathcal{N}(\mathbf{0}^{1:L}, \mathbf{I})$). For smooth transitions of sequences longer than the window length $L$, we incorporate the last $L'$ audio feature latents $\mathbf{w}_{s-L':0}$ from the preceding window as additional input. Thus the flow matching objective $\mathcal{L}_{\text{fm}}(\theta)$ is defined by

$$\mathcal{L}_{\text{fm}}(\theta) = \|\mathbf{v}_t^{-L':L}(\mathbf{x}_t, c_t; \theta) - \mathbf{u}_t(\mathbf{x}|\mathbf{w}_{s-L':L})\|, \quad (1)$$

where $\mathbf{x}_t = [\mathbf{w}_{s-L':L}|\varphi_t^{-L':L}(\mathbf{x}_0)] \in \mathbb{R}^{(-L'+L)\times d}$, $c_t \in \mathbb{R}^{(-L'+L)\times h}$ is the concatenated driving condition consisting of $[t, \mathbf{w}_{s-L':0}, \mathbf{w}_{s^{1:L}}]$. We also incorporate a velocity loss to supervise temporal consistency:

$$\mathcal{L}_{\text{v}}(\theta) = \|\Delta\mathbf{v}_t - \Delta\mathbf{u}_t\|, \quad (2)$$

where $\Delta\mathbf{v}_t$ and $\Delta\mathbf{u}_t$ are the frame-wise difference along the

time-axis for the prediction. The total objective $\mathcal{L}(\theta)$ is

$$\mathcal{L}(\theta) = \lambda_{\text{fm}}\mathcal{L}_{\text{fm}}(\theta) + \lambda_{\text{v}}\mathcal{L}_{\text{v}}(\theta), \quad (3)$$

where $\lambda_{\text{fm}}$ and $\lambda_{\text{v}}$ are coefficients. Additionally, we apply dropout to the preceding audio latents with a probability of $0.5$ to ensure a smoother transition in the initial window.

For the inference procedure, random waypoints are sampled from the source distribution and then flowed into the target audio latents by estimating the flow from $t = 0$ to $t = 1$ over steps. We could use multiple steps $1/\Delta t$ for inference:

$$\boldsymbol{x}_{t+\Delta t} = x_t + \Delta t\mathbf{v}_t^{1:L}, \qquad \text{for } t \in [0, 1] \quad (4)$$

**Robot Policy** We leverage the current and predicted audio frames to generate robot action with various manipulation policies. We adopt a flow matching policy proposed in [17], conditioned on the current audio, current image observation representation, and the future predicted audios. The robot policy is end-to-end trained in a supervised manner using the flow prediction MSE loss on future robot end-effector velocity of 16 steps.

**Learning Details** We decouple the audio flow matching world model from the robot policy training. Following the findings of [1], which investigates video generation and robotic control, a multi-stage training paradigm yields significantly higher performance—in terms of average success rate—than the end-to-end training. This modular architecture also allows independent adaptation and replacement of individual components. As described above, we employ AudioMAE for spectrogram encoding and reconstruction, and flow matching for the robot policy. In the subsequent experimental sections, we further demonstrate the flexibility of this framework in the piano-playing simulation task, where components can be interchanged—for instance, substituting AudioMAE with MusicVAE [18] for encoding and decoding Musical Instrument Digital Interface (MIDI) signals, and replacing the flow-matching robot policy module with a Soft Actor-Critic (SAC) reinforcement learning policy.

We use the ground truth future audio frames for training the robot policy. During evaluation, we use the generated audios from the latent world model to infer robot actions,

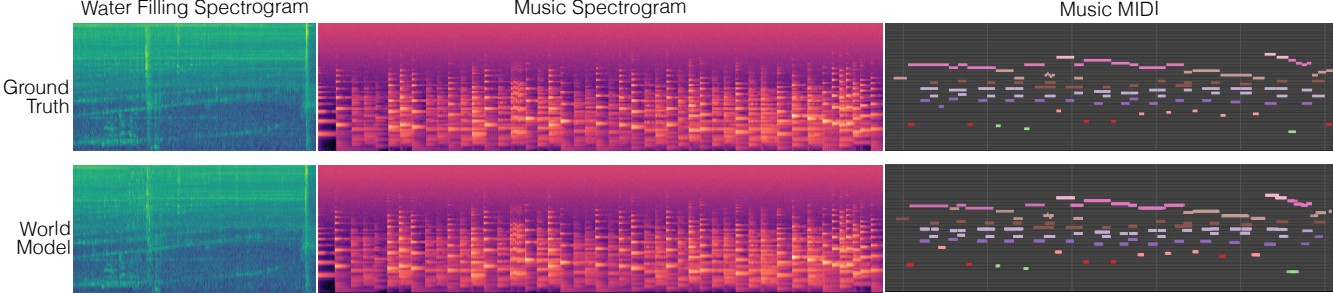

Fig. 2: Experimental results. We respectively show the ground truth and the world model generation of water filling spectrogram, music spectrogram, and MIDI data. The water filling spectrogram is predicted in a closed-loop manner during robot evaluation. Music pieces are generated autoregressively based on previous pieces.

based on the trained robot policy. We train with a batch size of 256 and AdamW with a learning rate of 1.5e-4. Learning rate is cosine decayed after 500 warmup steps, with 3000 training epochs. We incorporate a weight decay of 1e-6 on all trainable parameters. The overall model is relatively lightweight: training the entire pipeline—comprising the autoencoder, audio flow matching, and robot policy—on 20,000 spectrograms requires approximately one day on a single NVIDIA H100 GPU.

## IV. EXPERIMENTS

We conduct two robot experiments: 1) a real-world water filling task, and 2) a piano playing task in simulation.

### A. Filling Water

Our water-filling experiments are conducted using a Kinova Gen3 robotic arm, which operates the water dispenser by pressing and releasing its control button. RGB video data is captured using an Intel RealSense D410 camera mounted on the robot, while audio is recorded with a MAONO omnidirectional USB lapel microphone at 24-bit depth and a 192 kHz sampling rate. A Haption Virtuose 6D haptic device is employed to teleoperate the robot during data collection.

The audio waveform, sampled at 44.1 kHz, is transformed into a log Mel filterbank representation using the Kaldi implementation with a Hanning window, 128 Mel bins, and a frame shift of 10 ms. This process captures the perceptually relevant spectral features of the signal while suppressing noise through the use of Mel scaling and energy exclusion. The filterbank features are linearly normalized to range $[-1, 1]$. At each step, the last $128 \times 128$ spectrogram signals (roughly around 1.28 seconds) are used as the input to generate the future spectrogram with a size of $256 \times 128$ spectrogram (roughly around 2.56 seconds) using our proposed audio world model. Then we use the generated and observed spectrograms, together with a resized 224x224 resolution of synchronized image, to generate 6-DoF Cartesian space robot end-effector velocity commands with a horizon of 16 timesteps using the flow matching policy. The whole model prediction costs around 50 milliseconds at each step and operates in a closed-loop manner.

Fig. 2 shows the ground-truth and closed-loop generated spectrograms. The generated future data clearly captures the

onset, offset, and gradually increasing pitch of the activity. During testing, we also observed that the predicted button-releasing activity (the second yellow line shown in Fig. 2) consistently appeared in the generated spectrogram just before the bottle was actually filled. Our method achieved a 100% success rate over 30 trials, further demonstrating its capability to predict future audio states based on the underlying physical dynamics and pitch patterns.

### B. Piano Playing

In our second experiment, we simulate a piano duet scenario in which the robot must anticipate the forthcoming, yet-unheard musical notes from auditory input before performing its own part. We employ the simulated piano-playing environment introduced by [19]. Musical pieces are represented using the Musical Instrument Digital Interface (MIDI) standard, which encodes music as a sequence of time-stamped messages corresponding to note-on and note-off events. Following the approach in [19], we convert each MIDI file into a time-indexed note trajectory (also known as a piano roll), where each note is encoded as a one-hot vector. This trajectory serves as the agent's goal representation, specifying which keys should be pressed at each time step. We use a Soft-Actor-Critic (SAC) policy for robot joint action prediction at a frequency of 20 Hz.

As pointed in [19], to perform effectively, the agent must anticipate upcoming goals several seconds in advance; therefore, the goal state is stacked over a lookahead horizon. The goal state here consists of a vector of key goal positions obtained by indexing the piano roll at the current time step, along with a discrete vector indicating which fingers should be used. Instead of providing the SAC policy with the complete, predefined song as goal states, we employ our proposed audio world model to generate music dynamically, which then serves as the goal states for the RL agent. Our audio world model adopts MusicVAE to encode 64 time steps (around 8 seconds) of MIDI data and generate the next 64 steps. This process can be repeated autoregressively, using each newly generated segment as input to extend the sequence over time. The resulting MIDI is then converted into robot goal states for the SAC policy. We use the MIDI data from [19] and PIG [20] as our dataset. We train and

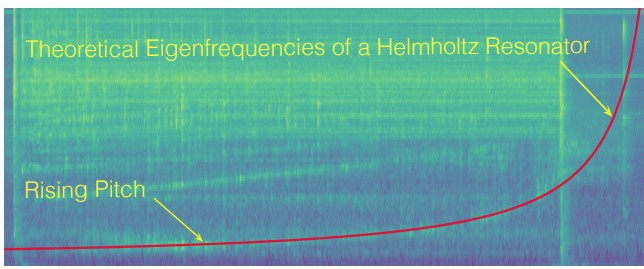

Fig. 3: Physics evaluation from world models. The latent spectral manifold generated by our model closely matches the theoretical eigenfrequencies of a Helmholtz resonator.

test on two songs (Twinkle Twinkle Little Star and Chopin Nocturne in E Flat Major Op.9).

We use the F1 score for evaluation, a widely adopted accuracy metric in the audio information retrieval literature. Our proposed world model–based learning framework, which incorporates generated future audio, is compared against a basic RL baseline without future lookahead. We observe improved performance when future goal states are incorporated into the observation. This result is intuitive, as access to future notes allows the policy to plan more effectively—for instance, by pre-positioning non-finger joints (e.g., the wrist) to reach upcoming notes more promptly. Fig. 2 shows, respectively, the ground-truth and autoregressively generated spectrograms, along with the corresponding MIDI data of the "Nocturne" piece.

### C. Physics Learning from World Models

To demonstrate that the generative world model provides a physically grounded representation of fluid-container interactions, we analyze the spectral evolution against the theoretical eigenfrequencies of a Helmholtz resonator. We define a unified state equation for the fundamental frequency $f$ over the temporal horizon $t \in [0, \tau]$ as:

$$f(t) = \mathcal{K} \left[ 1 - (1 - \phi) \left( \frac{t}{\tau} \right)^\alpha \right]^{-\frac{1}{2}} \quad (5)$$

where $\mathcal{K}$ represents the geometric acoustic constant, $\phi$ denotes the residual air volume ratio, and $\alpha = 1.7$ is the volumetric displacement exponent. As illustrated in Figure 3, the latent spectral manifold generated by our model tracks the theoretical trajectory with high fidelity. The precision of this alignment confirms that the world model has internalized the non-linear relationship between 3D volume displacement and acoustic resonance, suggesting an implicit understanding of the container's geometric constraints.

## V. Conclusions

We have proposed a latent flow matching–based audio generation model that facilitates robot policy learning. It is important to emphasize that this is not merely a multi-modal learning problem; rather, the audio cues in our tasks often capture intrinsic physical dynamics, such as pitch and rhythmic patterns. Consequently, constructing an audio-based world model is critical for informing robot action policies. Extending this framework to more complex tasks that require fine-grained and dexterous manipulation represents an important direction for future research.

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
