# OpenReview forum: "Learning Robot Manipulation from Audio World Models"
_IEEE.org/ICRA/2026/Workshop/Manipulation_Robustness — ICRA 2026_

### Official Review · Reviewer_63GY · 2026-05-06
**Interesting work on audio world model for robotic manipulation**

**Rating:** 6
**Confidence:** 4

**Review:**

Strengths:

- Well-motivated framing of audio as a primary modality, with water-filling and piano chosen as genuine cases where vision is degenerate.
- The latent flow-matching formulation is a lightweight architectural choice.
- The Helmholtz-resonator overlay (Fig. 3) is a creative qualitative probe suggesting the world model captures container acoustics rather than memorizing spectrograms.

Weaknesses:

- Missing key ablations. It trains the policy using ground-truth future audio while evaluating it with generated future audio. This introduces a potentially important train-test distribution gap for the policy, a common failure mode in world-model-conditioned control. A key ablation would be to train the policy on generated future audio, or compare against policies trained/tested with GT future audio, generated future audio, and no future lookahead.
- No quantitative baselines. Water-filling reports only 100% on 30 trials with no comparison against audio-without-future-prediction, MSE/diffusion prediction variants. Piano playing asserts "improved performance" with no F1 numbers, no variance. The "superior performance" claim is therefore unsupported.
- Mismatch with the workshop scope. For a venue on manipulation robustness under real-world challenges, there are no robustness perturbations: no background-noise injection, no microphone placement variation, no out-of-distribution bottles/liquids.

Overall:

The conceptual contribution — an audio-conditioned world model that anticipates contact-rich acoustic dynamics — is timely. But the gap between the strong claims and the actual evidence is large, and the fit with a robustness-themed workshop is weak. I would lean marginally accept; a revised version with quantitative baselines, an ablation of the GT-to-generated gap, and explicit acoustic robustness studies would be meaningful.

---

### Decision · Program_Chairs · 2026-05-21

Accept